# SQLGovernor: An LLM-powered SQL Toolkit for Real World Application

## Abstract

SQL queries in real-world analytical environments—whether written by humans or generated automatically—often suffer from syntax errors, inefficiency, or semantic misalignment, especially in complex OLAP scenarios. To address these challenges, we propose SQLGovernor, an LLM-powered SQL toolkit that unifies multiple functionalities—including syntax correction, query rewriting, query modification, and consistency verification—within a structured framework enhanced by knowledge management. SQLGovernor introduces a fragment-wise processing strategy to enable fine-grained rewriting and localized error correction, significantly reducing the cognitive load on the LLM. It further incorporates a hybrid self-learning mechanism guided by expert feedback, allowing the system to continuously improve through DBMS output analysis and rule validation. Experiments on benchmarks such as BIRD and BIRD-CRITIC, as well as industrial datasets, show that SQLGovernor consistently boosts the performance of base models by up to 10%, while minimizing reliance on manual expertise. Deployed in production environments, SQLGovernor demonstrates strong practical utility and effective performance.

## 1 Introduction

In real-world analytical applications, Structured Query Language (SQL) remains the primary interface for interacting with relational databases. Despite its maturity and widespread adoption, crafting accurate, efficient, and semantically aligned SQL queries—especially in complex analytical (OLAP) scenarios—remains a challenging task for both novice and experienced users alike.

OLAP workloads are central to modern business intelligence, reporting, and decision-making systems Codd (1993); Thomsen (2002). Even minor inefficiencies or ambiguities can lead to significant performance degradation, incorrect insights, or increased development overhead Vassiliadis & Sellis (1999); Pedersen & Jensen (2001). SQL queries in OLAP settings typically exhibit three key characteristics. First, they perform multi-dimensional analysis using advanced operations such as roll-up and drill-down Ceci et al. (2015), resulting in highly structured and deeply nested query forms. Second, these queries operate on large-scale enterprise data Chen et al. (2012), which increases computational costs and run-time unpredictability. Third, many OLAP queries are executed repeatedly—such as daily or weekly reports—making even small inefficiencies costly over time Zhan et al. (2019).

The repetitive and high-stakes nature of OLAP queries amplifies the need for robust, automated, and adaptive SQL post-processing solutions. Given these challenges, many tools have been developed to assist users in crafting better SQL queries, including syntax correction, query rewriting, and semantic refinement Cen et al. (2024); Chen et al. (2023); Liu & Mozafari (2024); Li et al. (2024c). While large language models (LLMs) have shown great promise in translating natural language questions into SQL, their applicability across the broader spectrum of SQL-related tasks remains underexplored. Moreover, in industrial practice, many users lack deep database expertise and often produce poorly written queries that are inefficient or semantically inaccurate, further exacerbating system performance and reliability issues Banisharif et al. (2022).

*C1: Productivity bottleneck from a fragmented ecosystem.* Existing SQL tools offer isolated functionalities, lacking a unified framework for tasks like syntax correction, semantic refinement, and

query rewriting Cen et al. (2024); Chen et al. (2023); Li et al. (2024c). This fragmentation creates a high barrier to entry for non-experts and increases manual effort by 30-40% for experienced practitioners due to context switching and compatibility issues team (2023a).

*C2: Lack of advanced techniques tailored for OLAP.* Most existing tools target general-purpose or simpler workloads like OLTP, NL2SQL, or offline optimization Chen et al. (2023); Wang et al. (2022); Cen et al. (2024); Yi et al. (2025). OLAP queries are complex, long-running, and require a careful balance between effectiveness and computational cost. Lightweight methods often fail to capture this complexity, while computationally intensive ones risk increasing end-to-end execution time Zhan et al. (2019).

*C3: High operational cost in an expert-centric knowledge lifecycle.* Correcting and rewriting queries demands deep domain and database expertise, which is difficult to capture with conventional data-driven methods. This reliance on expert teams increases labor costs by 25-35% Gartner (2022). Furthermore, maintaining and updating this expert knowledge is time-consuming, limiting scalability and adoption.

In summary, the current landscape lacks a comprehensive and practical SQL toolkit, which utilize evolving knowledge with fewer human efforts.

To address *C1*, we propose an LLM-powered SQL toolkit that unifies multiple functionalities within a structured framework enhanced by knowledge management. Users can either select individual tools for specific tasks or use an end-to-end pipeline that orchestrates multiple tools in a coordinated, use-case-driven manner. By consolidating diverse functionalities into a single platform, our approach eliminates the fragmentation in existing SQL tool-chains, significantly reducing deployment overhead, manual effort, and the barrier to entry.

To address *C2*, we adopt a dual approach that applies validated rules as guidance when appropriate, while permitting the LLM autonomous operation otherwise. Additionally, for particularly long and structurally complex queries, we propose a "fragment processing" strategy to reduce the chance of LLM hallucinations and lower the cost of using the LLM.

To address *C3*, we propose an expert-guided iterative self-learning mechanism to maintain a dynamic knowledge base for SQL tasks. The LLM agent analyzes DBMS outputs to generate new rules, identifying unseen error types from failed SQL and discovering rewriting strategies for inefficient queries. These rules are periodically verified by experts and integrated into the knowledge base for continuous improvement.

The main contributions of this paper can be summarized as follows:

1. Unified Framework: To the best of our knowledge, SQLGOVERNOR is the first comprehensive LLM-based SQL toolkit with a knowledge management module. It provides four core functionalities powered by a hybrid self-learning mechanism, thereby improving both user productivity and SQL quality.

2. Fragment Processing: We propose a fragment-wise processing strategy to address the complexity and length of OLAP queries. By localizing error detection and rewriting within individual fragments, including subqueries and CTEs, our approach enhances precision and reduces the cognitive burden on LLMs.

3. Hybrid Self-Learning: We introduce an expert-guided hybrid self-learning framework that enables SQLGOVERNOR to automatically extract common pattern from execution outputs, generate and validate new knowledge with minimal expert intervention, leading to continuous performance improvement.

4. Proven Effectiveness: Extensive experiments on academic benchmarks and real-world industrial datasets demonstrate that SQLGOVERNOR consistently improves the performance of base models by up to 10% in key metrics. Deployed in production environments, SQLGOVERNOR indicates strong practical utility and effective performance.

## 2 FRAMEWORK DESIGN

As illustrated in Figure 1, the architecture of SQLGOVERNOR comprises four specialized tools, each tailored to address a specific category of SQL-related tasks. These tools are supported by

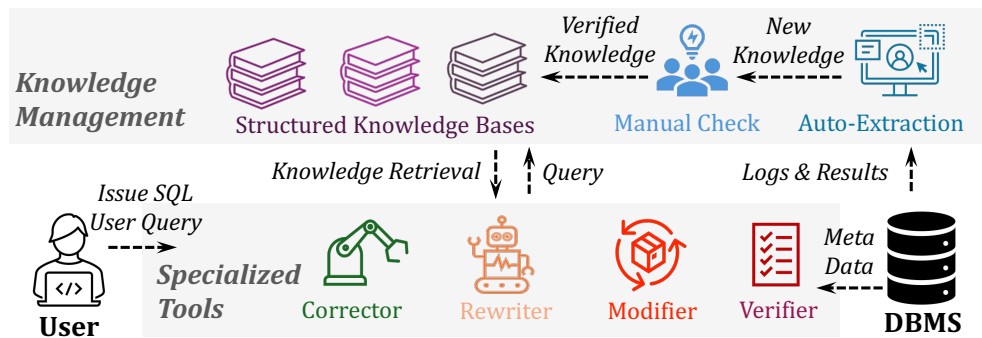

Figure 1: SQLGOVERNOR integrates four specialized SQL tools and a knowledge management module into a unified framework.

a knowledge management module that analyzes historical data and extracts actionable insights to guide error correction, query rewriting, and semantic refinement. We leave the detailed introduction to the knowledge management module in Appendix A.1.

Specifically, we focus on three common issues encountered during SQL execution in real-world applications: 1) resolving execution failures caused by syntax errors; 2) rewriting inefficient queries that result in excessively long execution times; 3) modifying SQL to better align with user intent.

## 3 SPECIALIZED SQL TOOLS

This section introduces our core design principle—the fragment processing strategy—which enables fine-grained, modular analysis for complex SQL tasks. Built upon this foundation, we further present four specialized tools in SQLGOVERNOR, each targeting a key subtask. The following subsections will detail three of these tools, while the elaboration of the equivalence verifier is provided in Appendix A.2.

### 3.1 FRAGMENT PROCESSING STRATEGY

Modern SQL queries, especially in OLAP workloads, often exhibit deeply nested structures with multiple layers of subqueries and Common Table Expressions (CTEs). To enable scalable and precise analysis, we propose a recursive fragment processing strategy that decomposes an input SQL query into smaller, self-contained fragments. Each fragment is analyzed independently under the same procedure, significantly reducing the reasoning complexity for LLM-based agents.

Concretely, the strategy operates as follows: the input query is first partitioned into a main query and a collection of CTEs. Each of these components is then recursively parsed to extract subqueries, which are likewise treated as fragments. For every fragment, the system performs a localized analysis and stores the result as a tuple of the form `<fragment_id, analysis_res>`. The pseudocode in Algorithm 1 outlines the core idea for fragment processing.

### 3.2 QUERY REWRITER

The Query Rewriter tool performs query optimization via a two-stage process: evaluation and rewriting. In the evaluation stage, the tool analyzes the input query to determine whether it exhibits inefficiencies. Specifically, it localizes the bottlenecks to specific fragments and proposes targeted rewriting strategies. In the rewriting stage, the tool applies the selected rewriting strategies to generate an optimized version of the original query. By leveraging the fragment-level structure produced during the fragment processing stage, the rewriting is both context-aware and fine-grained, ensuring that improvements are applied precisely without altering the query semantics.

**Evaluation** The evaluation process combines rule-based pattern matching with LLM-driven reasoning to deliver both precise and innovative optimization strategies—enhancing the overall efficiency of complex SQL queries.

If any rules match during the initial phase, their detailed descriptions are retrieved from the knowledge base and combined with the corresponding fragments into a structured prompt template (referred to as Scenario 1). If no rules are matched, the system evaluates the query against a set of "already efficient SQL" rules. A successful match indicates that the query is already optimal and does not require further rewriting efforts; otherwise, it suggests potential optimization opportunities that may not be covered by existing rules. In such cases, the query is passed to another predefined prompt template (referred to as Scenario 2).

Following the broad assessment, a detailed analysis is conducted using the LLM to refine and expand upon the rewriting suggestions. Specifically: (1) For Scenario 1, the prompt instructs the LLM to evaluate the applicability of each suggested rule, providing justifications and transforming general recommendations into actionable instructions where appropriate. (2) For Scenario 2 , the prompt directs the LLM to analyze the intent of the original query and explore alternative, more efficient formulations that preserve semantic equivalence. The output of this stage is standardized in `JSON` format, enabling seamless integration with the subsequent rewriting module.

To clarify the "fragment processing" design, Listing 1 provides a representative example. This SQL query contains six subqueries across different levels, with the deepest level of nesting being four. We have numbered all subqueries according to the order in which they are analyzed. The analysis results are as follows: Fragments 1-3 did not match any rules and meet the criteria for efficient SQL. Fragment 4 matched the `SAME_TABLE_JOIN` rule, as it was detected to scan the same table (e.g., `tb0`) twice. Fragments 5-6 also did not match any rules and are considered efficient. The outermost query (i.e., fragment 7) satisfies a rule involving a `LEFT JOIN` and an `IS NOT NULL` condition. These two defects were further confirmed by the LLM.

Listing 1: Show the working mechanism of the fragment processing design.

```
1  -- Fragment 7, Line 2-27
2  SELECT tb0.c0,
3  -- Fragment 5, Line 4
4  (SELECT tb3.c1 - tb3.c2 FROM tb3 WHERE tb3.ds = tb1.ds),
5  -- Fragment 6, Line 6
6  (SELECT AVG(tb4.c3) FROM tb4 WHERE tb4.ds = tb1.ds AND tb4.c3 > 100)
7  FROM tb1
8  LEFT JOIN tb2 ON tb1.ds = tb2.ds
9  WHERE tb2.ds is NOT NULL AND
10 tb1.dcrs <=
11 (
12   -- Fragment 4, Line 13-27
13   SELECT IFNULL(t1.c1 / t2.c2, 100) AS dcrs
14 FROM
15   -- Fragment 2, Line 16-22
16  (SELECT MIN(c) AS c1
17   FROM
18     -- Fragment 1, Line 19-22
19     (SELECT COUNT(*) AS c, ds
20      FROM tb0
21      WHERE ds >= '1014' AND ds < '1016'
22      GROUP BY ds)) AS t1,
23   -- Fragment 3, Line 24-26
24  (SELECT COUNT(*) AS c2
25   FROM tb0
26   WHERE ds = '1016') AS t2
27 )
```

**Rewriting**   As previously described, the Query Rewriter identifies potential inefficiencies in the input SQL and generates a set of actionable rewriting suggestions during the evaluation phase. In addition, the system retrieves relevant historical rewriting examples from the knowledge base by matching the current SQL fragments and their associated optimization rules. The LLM then integrates this information and synthesizes into a semantically equivalent yet execution-efficient SQL query that incorporates the suggested optimizations.

Listing 2 illustrates the rewritten SQL for the query presented in Section 5.2.1. Specifically, the pattern involving a `LEFT JOIN` combined with the `IS NOT NULL` condition is replaced with an `INNER JOIN`. This transformation is effective because an `INNER JOIN` naturally filters out rows with null values, thus achieving the same result as the original query but with a more efficient join operation. Furthermore, the two separate scans of table `tb0` are merged into a single scan to reduce

I/O overhead. The `WHERE` and `SELECT` clauses are appropriately adjusted to preserve the query's semantic correctness.

Listing 2: Show the rewritten SQL for the example in Listing 1.

```
1  WITH cte AS
2  (SELECT
3      IFNULL(MIN(CASE WHEN ds >= '1014' AND ds < '1016' THEN cnt END) /
4              SUM(CASE WHEN ds = '1016' THEN 1 ELSE 0 END), 100) AS dcrs
5  FROM (
6      SELECT ds, COUNT(*) AS cnt
7      FROM tb0
8      WHERE ds >= '1014' AND ds <= '1016'
9      GROUP BY ds
10 )
11 SELECT tb0.c0,
12 (SELECT tb3.c1 - tb3.c2 FROM tb3 WHERE tb3.ds = tb1.ds),
13 (SELECT AVG(tb4.c3) FROM tb4 WHERE tb4.ds = tb1.ds AND tb4.c3 > 100)
14 FROM tb1
15 INNER JOIN tb2 ON tb1.ds = tb2.ds
16 WHERE tb1.dcrs <= (SELECT dcrs FROM cte)
```

## 3.3 QUERY MODIFIER

We classify modification requests into four general categories: (1) *Realizing a specified semantics*: Adjust the SQL logic to align with the user's intended meaning. (2) *Explaining the SQL*: Preserve the original logic while adding comments or annotations for clarity. (3) *Adopting a specified syntax*: Maintain semantic equivalence while adapting the query to match the user's stylistic or structural preferences. (4) *Other SQL-related tasks*: Capture queries that do not clearly fall into the above three categories, such as stylish polishing. To fulfill each request, the tool follows a three-step pipeline: (1) metadata preparation, (2) user intent clarification, and (3) query modification. Note that the definitions of categories are rather flexible and can be customized.

**Metadata Preparation** During the metadata preparation stage, we extract the target SQL snippet along with its surrounding context to provide a comprehensive view of the query environment. We gather metadata from two primary sources. The first includes tables and columns referenced in the SQL snippet, along with their names and descriptions. The second source is derived from the user's historical query logs, where we identify the top-$k$ most frequently accessed tables. For each of these tables, we extract relevant metadata—such as schema information and usage patterns—to help the LLM better understand the context and semantics of the query. Additionally, we append a current timestamp to the metadata to provide temporal grounding, which is particularly useful when handling evolving schema or time-sensitive queries.

**User Intent Clarification** The user intent clarification step maps the natural language request to one of the four predefined categories described earlier. This classification combines two complementary strategies: heuristic keyword matching and semantic similarity scoring. In the heuristic keyword matching strategy, we identify a set of domain-specific keywords and phrases that are commonly associated with each modification type. For each category $C_j$, we define a corresponding keyword set $\mathcal{KW}_j = \{k_{j1}, k_{12}, \ldots, k_{jn_j}\}$. Given a user request $\mathcal{Q}$, we compute a weighted matching score $S_j^{kw}$ for each category as $S_j^{kw} = \frac{1}{N_j} \sum_{k_{ji} \in \mathcal{KW}_j} \mathrm{match}(\mathcal{Q}, k_{ji}) \times w_{ji}$, where $N_j$ is the total number of candidate keywords in $\mathcal{KW}_j$; $\mathrm{match}(\mathcal{Q}, k_{ji})$ is a binary function returning 1 if the keyword $k_{ji}$ appears in the request $\mathcal{Q}$, and 0 otherwise; $w_{ji}$ denotes the weight assigned to keyword $k_{ji}$, reflecting its relative importance within category $C_j$.

To complement the keyword-based method, we also employ semantic embeddings to capture more nuanced intent signals. We have explored two distinct embedding pathways: (1) Sentence-Transformer with Masking: We pre-process the query by replacing specific metadata (e.g., table/-column names) and constant values with special tokens `[MASK]`, then encode the masked text into a vector $\mathbf{e}_{\mathcal{Q}}$ using a Sentence-Transformer model (e.g., RoBERTa Liu et al. (2019)). (2) Instruction-aware Qwen3-Embedding: We utilize the Qwen3 embedding model Zhang et al. (2025) to encode the original query $\mathcal{Q}$, guided by an instruction prompt that directs the model to focus on the user's action intent rather than details such as schema identifiers. During our development and testing, the Instruction-Aware Qwen3 Embedding consistently outperforms the masking-based

Sentence-Transformer approach in both classification accuracy and robustness to domain variations. This is attributed to its ability to better align with the LLM's internal reasoning process and its use of instruction-tuned representations that emphasize action-oriented semantics. What's more, real-world user queries often suffer from ambiguity, incomplete descriptions, or informal phrasing. In such cases, rule-based detection and masking strategies tend to fail.

Once the embedding method has been selected, we construct a representative embedding vector $\mathbf{e}_{\mathcal{C}_j}$ for each category $C_j$. These are derived by collecting historical user queries, classifying their intents using an LLM, and computing the centroid embedding for each category.

We employ cosine similarity between the query embedding $\mathbf{e}_Q$ and the category centroid embeddings $\mathbf{e}_{\mathcal{C}_j}$ as a measure of semantic proximity. This semantic score is combined with the heuristic keyword matching score to form the final classification decision. Formally, we define the final classification score $F_j$ for each category $\mathcal{C}_j$ as $F_j = \alpha \cdot S_j^{kw} + \beta \cdot \text{similarity}(\mathbf{e}_Q, \mathbf{e}_{\mathcal{C}_j})$, where $\alpha$ and $\beta$ are weighting parameters used to balance the heuristic keyword matching score and the semantic similarity score. To ensure robust and reliable intent clarification, we introduce a confidence threshold $\theta$. If the maximum classification score $\max(F_j)$ across all categories falls below this threshold, the system treats the request as unsupported and rejects the modification task. This mechanism helps filter out ambiguous or outlier queries that do not align well with any known modification type, thereby maintaining the integrity and reliability of the classification pipeline.

**Modification** Once the necessary data has been collected and the user's intent has been classified, we construct a structured prompt tailored to the identified modification type. The prompt integrates the following components: the original SQL fragment, its surrounding context, relevant metadata (e.g., schema information and top accessed tables), the current timestamp for temporal grounding, and the natural language instruction from the user. This contextualized prompt is then fed into the LLM, which reasons over the input and generates a modified SQL fragment that accurately fulfills the user's intent while preserving correctness and consistency.

## 3.4 SYNTAX ERROR CORRECTOR

The overall syntax correction workflow comprises three key stages: clarification, data preparation, and correction.

**Clarification** The clarification stage begins by extracting structured information—such as the exception type, error location, and descriptive message—from DBMS error logs. This data is used for embedding-based retrieval against a knowledge base of known error patterns and correction strategies. Each retrieved strategy provides three key components to guide the LLM: (1) Schema Dependency: Indicates whether the error requires access to schema metadata (e.g., for column-related errors) to avoid including large, complex schemas unnecessarily. (2) Correction Scope: Classifies the error as either localized (e.g., a missing comma) or global, helping to focus the LLM's attention. (3) Correction Hints: Provides explicit, actionable guidance for the LLM, such as explaining the root cause of a `Column count mismatch` error. This approach of selective schema inclusion and localized correction keeps the prompt concise. This not only reduces inference latency and cost but also prevents irrelevant information from interfering with the model's reasoning capabilities.

**Data Preparation** Guided by the outputs from the clarification stage, the data preparation step determines what information should be included in the final prompt. It selectively extracts relevant schema components, or isolates specific query fragments for localized correction, depending on the retrieved strategy. If the clarification stage fails to find a confident match in the knowledge base, a conservative fallback strategy is applied: the full schema is retained, and a global correction approach is used.

**Correction** In the final correction stage, the erroneous SQL (or fragment), along with its associated context—including selectively extracted schema information and tailored guidance—is assembled into a structured prompt. The LLM agent then processes this input and generates a corrected version of the SQL query.

# 4 EXPERIMENTS

This section presents the main results of our experiments on the BIRD and BIRD-CRITIC benchmarks, along with a partial ablation study. For more comprehensive details on the datasets, experimental settings, and additional results, please refer to the Appendix A.4- A.6.

## 4.1 MAIN RESULTS

**Results on BIRD**  As SQLGOVERNOR is used as post-processing tool for the NL2SQL task, we select three strong base models and one representative baseline. The base models are CodeS-7B, CodeS-15B Li et al. (2024a) and XiYanSQL Gao et al. (2024b) and the baseline is *SQLFixAgent* Cen et al. (2024).

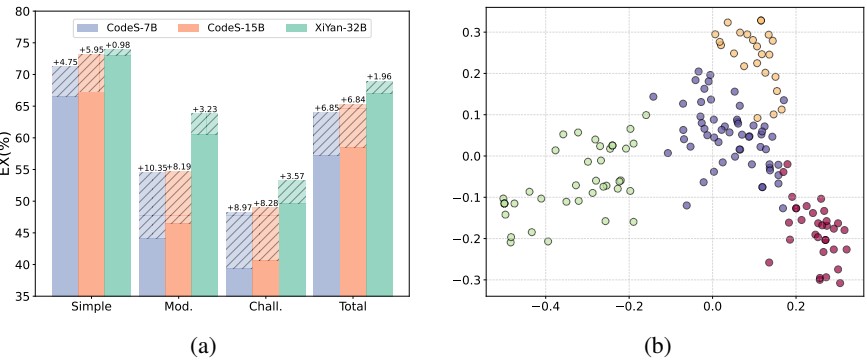

Figure 2: (a) Performance on various difficulty data categories. (b) Visualization results of user query clustering.

Table 4 presents a detailed comparison of SQLGOVERNOR's performance against existing methods on the BIRD dataset. When integrated with CodeS-15B, SQLGOVERNOR achieves EX and VES scores of 65.32% and 67.87%, respectively, representing improvements of 6.84% in EX and 8.00% in VES over the baseline. These gains outpace those of SQLFixAgent, which improves the baseline by only 3% in EX and 4.35% in VES, highlighting SQLGOVERNOR's superior effectiveness. When paired with XiYan-32B, SQLGOVERNOR further enhances the already high baseline scores of 68.97% (EX) and 70.89% (VES), achieving marginal but meaningful improvements of 1.96% in EX and 3.10% in VES.

Furthermore, we carefully analyze the detailed performance of SQLGOVERNOR across various base models. Figure 2a presents the results of this method on the three difficulty levels (`Simple`, `Mod.`, and `Chall.`) of the BIRD dev set. The solid bars represent the results of the base model, while the dashed bars above indicate the gains achieved by SQLGOVERNOR. It is evident that SQLGOVERNOR achieves significantly higher gains on the `Mod.` and `Chall.` difficulty levels compared to Simple. Notably, on the CodeS-15B and XiYan-32B models, the metric gains for `Chall.` even surpass those for Mod., making it the highest-performing category among the three difficulty levels, with respective gains of +8.28% and +3.57%. This clearly demonstrates the advantage of SQLGOVERNOR in handling long and complex SQL queries.

In summary, SQLGOVERNOR exhibits strong performance improvements across all tested baseline models, outperforming alternative methods such as SQLFixAgent and demonstrating promising value even when applied to state-of-the-art models like XiYan-32B.

Table 1: Evaluation of SQLGOVERNOR on BIRD-CRITIC-Flash. Metric: SR (%).

| Method | | Category | | | | | | | | |
|--------|--|----------|--|------------|--|-----------------|--|------------|--|-------|--|
| | | Query | | Management | | Personalization | | Efficiency | | Total | |
| **Qwen3-32B** | / | 18.8 | / | 34.0 | / | 28.1 | / | 22.7 | / | 26.0 | / |
| | + SQLGOVERNOR | 21.9 | ↑3.1 | **54.0** | ↑**20.0** | 35.9 | ↑7.8 | 36.4 | ↑**13.7** | 36.0 | ↑**10.0** |
| **Qwen2.5 -72B-Instruct** | / | 20.3 | / | 46.0 | / | 32.8 | / | 31.8 | / | 32.0 | / |
| | + SQLGOVERNOR | **26.6** | ↑**6.3** | 52.0 | ↑6.0 | **43.8** | ↑**11.0** | **45.5** | ↑13.6 | **40.5** | ↑8.5 |

**Results on BIRD-CRITIC-Flash**  To evaluate the effectiveness of SQLGOVERNOR in addressing SQL issues arising from user-provided natural language queries, we conduct experiments on the

BIRD-CRITIC-Flash benchmark. Our approach dynamically routes each SQL issue to the most suitable tool: the Query Rewriter for efficiency-related problems, the Syntax Error Corrector for execution errors, and the Query Modifier for other semantic or stylistic adjustments.

We evaluate two widely used LLMs—Qwen3-32B and Qwen2.5-72B-Instruct—in both the original configurations provided by the benchmark team and with our toolkit integrated. As shown in Table 1, SQLGOVERNOR consistently improves performance across all categories and both base models.

For Qwen3-32B, integrating SQLGOVERNOR leads to substantial gains, particularly in the Management category (+20.0%), where the success rate increases from 34.0% to 54.0%. Significant improvements are also observed in Efficiency (+13.7%) and Personalization (+7.8%), indicating that both query rewriting and semantic alignment benefit greatly from our toolkit. Overall, the total score rises from 26.0% to 36.0%. When applied to Qwen2.5-72B-Instruct, SQLGOVERNOR still delivers consistent gains. The largest improvement is seen in Personalization (+11.0%).

In addition to success rate, we measure the average end-to-end inference time per query on both models. For Qwen3-32B, the runtime increases from 8.5s (base) to 18.4s with our toolkit; similarly, for Qwen2.5-72B-Instruct, it increases from 9.8s (base) to 21.3s. While this represents a non-trivial overhead, it is largely due to the multi-stage processing pipeline—including intricate problem analysis and solving process powered by LLM-that is essential for achieving high-quality corrections in complex OLAP queries.

## 4.2 ABLATION STUDY

**User Intent Clarification**  To evaluate the effectiveness of user intent clarification in the SQL Modifier, we sampled 150 real-world query tasks from our production environment and manually annotated them with intent categories. We then tested the performance of the Instruction-Aware Qwen3 Embedding in classifying these intents. Specifically, we used an embedding model with 8B parameters and a vector dimension of 1024. As a baseline, we also evaluated Qwen3-32B, where the LLM directly performs classification without prior embedding-based filtering. Both models were deployed under identical execution environments to ensure fair comparison.

The results are as follows: the Instruction-Aware Qwen3 Embedding achieves an accuracy of 78.9%, with an average inference latency of 0.173 seconds. In contrast, Qwen3-32B achieves higher accuracy at 84.3%, but incurs a significantly higher average latency of 0.354 seconds. These findings suggest that while the LLM-based classifier offers relatively higher accuracy (5.4%), the embedding-based approach provides a favorable trade-off between speed and performance—making it particularly suitable for high-throughput or latency-sensitive applications.

To provide a more intuitive understanding of the embedding quality, we applied PCA to reduce the embedding vectors to two dimensions and visualized them using scatter plots, as shown in Figure 2b.

**Rewriting**  To evaluate the performance of SQL rewriting tool, we used Payment-SQL, a test set that closely aligns with OLAP scenario, and employed ETS and ETOG as evaluation metrics. We selected four representative models as baselines: Qwen2.5-72B-Instruct Hui et al. (2024), Qwen3-32B Zhang et al. (2025), LLM-R$^2$ Li et al. (2024c), and GenRewrite Liu & Mozafari (2024). Specifically, Qwen2.5-72B-Instruct and Qwen3-32B are general-purpose LLMs that are instructed to rewrite the input SQL query in a single inference step. LLM-R$^2$ employs LLMs to select appropriate rewriting rules and trains a separate demonstration recommendation model to guide the rewriting process. GenRewrite represents the first non-rule-based, end-to-end query rewriting approach that fully leverages the generative capabilities of LLMs. To ensure fair comparisons, we maintained identical execution environments for the SQL queries before and after rewriting during testing.

Table 2: Execution efficiency of SQLs in Payment-SQL after rewriting and time-cost for rewriting.

| Methods | ETOG(%) | ETS(s) | Cost(s) | $\Delta$(s) |
|---|---|---|---|---|
| Qwen3 | 11.06 | 20.19 | 15.24 | ↑4.95 |
| Qwen2.5 | 14.56 | 26.59 | 18.41 | ↑8.18 |
| LLM-R$^2$ | 29.87 | 54.53 | 46.85 | ↑7.68 |
| GenRewrite | 31.25 | 57.07 | 38.36 | ↑18.71 |
| SQLGOVERNOR | **45.92** | **83.86** | 30.73 | **↑53.13** |

The results are presented in Table 2. From the table, we observe that SQLGOVERNOR exhibits a significant performance advantage when applied to industrial-level OLAP workloads. On average, across the entire test set, SQLGOVERNOR achieves a 45.92% reduction in execution time and an 83.86-second reduction in absolute execution time. We also report the rewriting cost, i.e., the time required to perform the rewriting itself, and the net benefit ($\Delta$), defined as the difference between ETS and Cost. Notably, while SQLGOVERNOR incurs a relatively moderate rewriting overhead (30.73s), it delivers the largest net benefit (+53.13s), demonstrating its practical viability in real-world applications where query latency is critical.

Listing 3 presents an example of an rewritten SQL query from Payment-SQL. During the evaluation stage, the LLM provided the following rewriting suggestions: (1) Use the `WITH` clause to explicitly define the result of `UNION ALL` as a temporary table, making it easier for the rewriter to understand and optimize the query. (2) In the `UNION ALL` step, explicitly select only the columns that are actually needed, avoiding the retrieval and processing of unnecessary data.

Listing 3: Example of rewriting result from Payment-SQL.

```sql
-- Original SQL
SELECT AVG(duration)
FROM (
    SELECT *, row_number() OVER (PARTITION by instanceid ORDER BY modifytime
        DESC) AS id
    FROM (
        SELECT *
        FROM table0
        WHERE ds > '0201'
        UNION ALL
        SELECT *
        FROM table1
        WHERE ds > '0201'
    ) a
) b
WHERE id = 1 AND taskid IN (1, 12, 123) AND scriptid = 666
-- Rewritten SQL
WITH combined_data AS (
    SELECT taskid, instanceid, scriptid, modifytime
    FROM table0
    WHERE ds > '0201'
    UNION ALL
    SELECT taskid, instanceid, scriptid, modifytime
    FROM table1
    WHERE ds > '0201'
)
SELECT AVG(duration)
FROM
(
    SELECT *, ROW_NUMBER() OVER (PARTITION BY instanceid ORDER BY modifytime
        DESC) AS id
    FROM combined_data
) b
WHERE id = 1 AND taskid IN (1, 12, 123) AND scriptid = 666
```

## 5 CONCLUSIONS

In this work, we present SQLGOVERNOR, the first comprehensive LLM-based SQL toolkit with integrated knowledge management. It unifies four core functionalities—syntax correction, query rewriting, semantic refinement, and consistency verification—into a single framework powered by a hybrid self-learning mechanism.

One of the key innovations lies in its fragment-wise processing strategy. By focusing on individual fragments such as subqueries and CTEs, the approach improves precision while reducing the cognitive burden on LLMs. Moreover, SQLGOVERNOR incorporates an expert-guided hybrid self-learning framework that continuously enhances performance by extracting patterns from execution outputs and validating generated rules with minimal expert input.

Extensive experiments show that SQLGOVERNOR consistently boosts base models' performance by up to 10% in key metrics on benchmarks like BIRD and BIRD-CRITIC. Deployed in production environments, it demonstrates strong utility and adaptability across real-world databases.

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

# A APPENDIX

## A.1 KNOWLEDGE MANAGEMENT

To enhance the effectiveness of LLMs in SQL-related tasks, we design a knowledge management module paired with dedicated recall and retrieving techniques. SQLGOVERNOR benefits from past experiences and domain-specific insights to reduce redundant computations and improve both accuracy and consistency. To further automate knowledge acquisition and minimize reliance on manual expert input, we implement structured knowledge bases inspired by the self-reinforcing data flywheel mechanism Roiter (2025). Each knowledge base continuously accumulates and updates rules from both successful and failure cases, enabling the system to improve over time with minimal supervision.

| Index | Description |
|---|---|
| COUNT(DISTINCT)_SAME_FIELD | Pre-aggregate using a CTE and convert complex condition into 0/1 flags with the `MAX` function and then use `SUM` to count these flags. |
| COUNT(DISTINCT)_DIFFERENT_FIELD | Decompose multiple `COUNT(DISTINCT)` expressions within the same `SELECT` clause into separate subqueries, each computing a single `COUNT(DISTINCT)` to enable parallel execution. Subsequently, the results are combined using `JOIN` or `UNION ALL` operations. |
| IN(SELECT) | Rewrite the `IN (SELECT xxx)` clause as a `JOIN (SELECT xxx)` operation and apply the `MAPJOIN` hint to fully load the subquery's corresponding small table into memory. |

(a) "Rules"

| Index | Details | Tag |
|---|---|---|
| Embedding of the raw SQL's template | Raw SQL:
SELECT * FROM table1 WHERE id IN (SELECT id FROM table2 WHERE cond = 0)
Rewritten SQL:
SELECT /*+ MAPJOIN(t2) */ t1.* FROM table1 t1
JOIN (SELECT DISTINCT id FROM table2 WHERE cond = 0) t2 ON t1.id = t2.id | IN(SELECT) |
| Embedding of the raw SQL's template | Raw SQL:
SELECT ds, COUNT(DISTINCT filed_1) as unique_filed_1,
COUNT(DISTINCT filed_2) as unique_filed_2, COUNT(DISTINCT filed_3) as unique_filed_3
FROM table GROUP BY ds
Rewritten SQL:
WITH filed_1_count AS (SELECT ds, COUNT(DISTINCT filed_1) as unique_filed_1
FROM table GROUP BY ds),
filed_2_count AS (SELECT ds, COUNT(DISTINCT filed_2) as unique_filed_2
FROM table GROUP BY ds),
filed_3_count AS (SELECT ds, COUNT(DISTINCT filed_3) as unique_filed_3
FROM table GROUP BY ds)
SELECT u.ds, u.unique_filed_1,  o.unique_filed_2, p.unique_filed_3
FROM filed_1_count u JOIN filed_2_count o ON u.ds = o.ds JOIN filed_3_count p ON u.ds = p.ds | COUNT(DISTINCT)_SAME_FIELD |

(b) "Historical Data"

Figure 3: Demonstration of the knowledge base serving the Query Rewriter tool.

**Structure of the Knowledge Base** The knowledge base is carefully structured to meet the diverse needs of different SQL tools, with each tool having its own dedicated repository. Each repository is divided into two sub-modules: "Rules" and "Historical Data". The "Rules" sub-module contains structured entries that provide actionable guidance for the LLM in addressing specific SQL-related tasks. These rules include transformation patterns, rewriting strategies, and syntactic corrections, enabling the model to apply well-defined solutions in a consistent manner. The "Historical Data" sub-module stores high-quality, representative cases collected from real-world applications of the SQL tools. By analyzing these past examples, the LLM can recognize recurring patterns and adopt strategies that have proven effective in similar contexts.

Figure 3 illustrates the structure of the knowledge base used by the Query Rewriter. Each rule entry consists of two fields: *index*, used for efficient retrieval, and *description*, which provides a detailed

explanation of the rule's application and scope. Each historical data entry includes three fields: *index* (for retrieval), *details* (a full description of the case), and *tag*, which links the case to relevant rule indices, thereby facilitating cross-referencing between observed problems and applicable solutions.

**Knowledge Storage and Retrieval**    To effectively support the diverse types of knowledge entries, we design tailored storage and retrieval strategies for each sub-module of the knowledge base. For the "Historical Data" module, all entries are stored in a vector database—specifically Star-Rocks team (2023b). Retrieval is based on the structural and semantic similarity between SQL templates. To facilitate this, we first extract the template of each SQL query by replacing concrete identifiers (e.g., table and column names) with symbolic placeholders. This abstraction allows the encoder to focus on high-level patterns rather than surface-level variations. During retrieval, the input SQL is similarly transformed into a template, encoded into a vector representation, and used to retrieve the top-$k$ most similar historical cases via cosine similarity. These candidates are further filtered using the *Tag* field to ensure alignment with the specific task or error type.

In contrast, the "Rules" module employs different strategies depending on its application context. For the Query Rewriter, each rule is associated with a unique label (e.g., `IN(SELECT)`) that captures its applicability condition. We store these rules in an ElasticSearch Elasticsearch (2018) database to enable efficient exact matching during retrieval. For the Syntax Error Corrector, exception categories (e.g., `RuntimeException`, `SqlValidatorException`) are often too coarse-grained to be informative. Therefore, we retain more detailed error messages (e.g., `SqlValidatorException: INNER, LEFT, RIGHT or FULL join requires a condition (NATURAL keyword or ON or USING clause)`) as retrieval targets. These messages are stored in a vector database. Given the verbosity of real-world SQL execution logs, we first apply regular expressions to extract key information before encoding it into vector representations for retrieval.

**Hybrid Self-Learning Mechanism**    To ensure the knowledge base is both effective and actionable, we adopt a multi-source initialization strategy tailored to each sub-module. For the "Rules" sub-module, initialization involves aggregating knowledge from diverse sources and organizing it into structured entries. For the Query Rewriter, the rule set is primarily sourced from domain experts and is designed to complement the built-in optimization rules of the DBMS. These expert-defined rules have been rigorously validated to ensure their practical effectiveness in real-world OLAP scenarios. For the Syntax Error Corrector, the rule base is initialized using frequently asked questions (FAQs) and common error-handling guidelines from technical documentation. These are formalized into ¡exception, fixing action¿ pairs and further mapped to the ¡index, description¿ structure for retrieval compatibility. For the "Historical Data" sub-module, initialization is conducted in two ways: (1) manually curating high-quality cases that align with existing rules, and (2) extracting representative queries from execution logs. These cases are selected based on criteria such as execution frequency, complexity, and historical performance, ensuring their relevance and utility in future inference tasks.

Furthermore, both sub-modules support incremental updates through the self-learning mechanism, allowing the system to refine its knowledge over time based on new data and user feedback. Specifically, we design a rule update paradigm that automatically extracts supplementary rules by analyzing SQL queries that fail to meet user requirements—such as those resulting in execution errors, returning incorrect results, or exhibiting excessive runtime. A heuristic pattern recognition-based data filtering algorithm is first applied to extract relevant features (e.g., error messages, query structures, execution times) from execution logs. An LLM agent then analyzes this information, identifies common inefficiencies or mistakes, and generates structured knowledge entries. The prompt used for rule generation is presented in Listing 4.

Listing 4: Prompts for generating rules.

```
Task Description:
    You are provided with an SQL query along with its execution outputs (e.g. logs and
        results) from DBMS. Your task is to analyze the logs and results to identify
        potential errors or inefficiencies in the query.
    -
Instruction:
    1. Review the execution logs and results to determine whether the query contains errors
        or inefficiencies.
    2. For each confirmed problem, try to distill it into a generalizable rule, including the
        abstract problem pattern, detailed description, and its solution.
```

```
    3. Convert your findings into JSON format, where the key is the problem pattern used as
        index and the value is a detailed description of the problem along with possible
        corrective actions.
    -
Demonstration:
    {Few-shot examples curated from previously validated rule entries.}
    -
Question:
    {SQL query and user query if necessary.}
    -
Execution Outputs:
    {From logs and results.}
```

To ensure accuracy and reliability, experts verify the newly generated rules based on predefined conditions. We implement a threshold-triggered mechanism to maintain the relevance of the knowledge base. Verification is triggered when either: (a) the number of new rules exceeds a threshold $t_1$, or (b) the time elapsed since the last update surpasses a threshold $t_2$, where $t_1 = \lfloor \lambda \cdot \sqrt{N_{\text{current}}} \rfloor$ and $t_2 = \beta \cdot \mathbb{E}[\Delta t_{\text{historical}}]$, with $\lambda = 2.5$ controlling capacity scaling and $\beta = 1.3$ for temporal adaptation. Here $N_{\text{current}}$ denotes the current number of rules, and $\mathbb{E}[\Delta t_{\text{historical}}]$ represents the expected historical update interval. To avoid redundancy, semantically equivalent rules are identified and clustered through the following process: (a) The description field of rule $r_i$ is encoded using RoBERTa-base Liu et al. (2019). (b) Compute the pairwise similarity. (c) Rules are grouped around $r_i$ into $C_k(i)$ based on the similarity score using DBSCAN Schubert et al. (2017). (d) Merge semantically equivalent rules using centroid synthesis.

$$\mathbf{e}_i = \text{RoBERTa}(r_i.\text{get}(\text{description})) \in \mathbb{R}^{768} \tag{1}$$

$$s(r_i, r_j) = \frac{\mathbf{e}_i^\top \mathbf{e}_j}{\|\mathbf{e}_i\|\|\mathbf{e}_j\|} \tag{2}$$

$$r_{\text{new}} = \arg \min_{r \in C_k(i)} \sum_{r_i \in C_k(i)} \|\mathbf{e}_r - \mathbf{e}_{r_i}\|_2 \tag{3}$$

The updated rules are then applied to subsequent tasks, generating new data that perpetuates the improvement cycle.

The update mechanism for the "Historical Data" sub-module is relatively straightforward. As mentioned earlier, for rules that have been validated by experts, we incorporate the corresponding instances into the historical data. This continuous cycle of knowledge collection, validation, and application ensures that the knowledge base remains up-to-date and effective, thereby enhancing the overall performance of the LLM-based SQL toolkit.

### A.2 EQUIVALENCE VERIFIER

We conduct a systematic review of representative SQL equivalence verification methods and summarize their characteristics in Table 3.

Formal methods such as SPES Zhou et al. (2022) offer rigorous correctness guarantees through symbolic execution but are limited in practical applicability due to type constraints and poor performance on real-world benchmarks like TPC-H TPC. Graph-based approaches (e.g., FuncEvalGMN Zhan et al. (2025)) achieve broader coverage via structural matching but require extensive model training, leading to high deployment costs and limited generalization across diverse query patterns. LLM-based solutions Zhao et al. (2023) employ advanced prompting techniques but still suffer from limited accuracy on complex queries and exhibit positive bias in equivalence judgments Wu et al. (2024).

Table 3: Comparison of representative SQL equivalence verification methods.

| Method | Technical Basis | Correctness Guarantee | Applicable Scope | Deployment Cost |
|---|---|---|---|---|
| SPES Zhou et al. (2022) | Symbolic Execution | ✓ | Very Limited | Low |
| SQLSolver Ding et al. (2023a) | Formal Logic | ✓ | Limited | Low |
| FuncEvalGMN Zhan et al. (2025) | Graph Matching | ✗ | General | High |
| LLM-SQL-Solver Zhao et al. (2023) | Probabilistic LM | ✗ | General | Medium |
| SQLGOVERNOR | Probabilistic LM | ✗ | SELECT-based DML | Medium |

To address these limitations, we propose a structured framework for SQL semantic equivalence verification, specifically tailored for SELECT-based DML queries in OLAP scenarios. Our approach

decomposes the verification process into two stages: (1) semantic intent extraction, which captures the meaning of each query using controlled LLM-mediated interpretation, and (2) hierarchical consistency checking, which performs field-level alignment and derivation trace analysis to assess equivalence.

In the first stage, each SQL query is translated into a structured natural language representation that captures the provenance of every field in the main `SELECT` clause. This includes source tables, transformation logic (e.g., aggregations or arithmetic expressions), and filtering or joining conditions. To enhance interpretability, we adopt a stepwise parsing strategy that starts with the innermost subqueries and progressively builds up the outer query semantics.

A lightweight pre-filtering module eliminates clearly inconsistent query pairs based on schema-level heuristics, such as mismatches in field count or source tables. For remaining pairs, an LLM agent performs detailed consistency analysis by aligning corresponding `SELECT` fields and reasoning about semantic equivalence.

Our prompt design enforces bidirectional field mapping, supports counterexample generation for non-equivalent pairs, and incorporates calibrated confidence scoring—ensuring both the interpretability and reliability of the verification process.

## A.3 RELATED WORK

**Query Rewriting** Query rewriting can be classified according to the stages of the SQL query lifecycle. A typical query goes through several phases: parsing for syntax validation, binding for schema resolution, optimization using a cost model, and execution by the database engine. Based on this lifecycle, rewriting can be applied at different levels: (1) raw SQL queries before parsing, (2) logical plans generated by the binder, or (3) physical plans produced by the optimizer within the DBMS.

Physical plan rewriting focuses on selecting the most efficient execution plan among functionally equivalent alternatives. The DBMS optimizer employs search algorithms—such as dynamic programming Graefe (1995); Graefe & McKenna (1993)—to explore the space of physical plans without exhaustive enumeration. A cost model is used to estimate the execution cost of each candidate plan. Recent approaches have introduced machine learning and deep learning techniques Kipf et al. (2018); Trummer et al. (2021); Sun & Li (2019) to enhance cost estimation and plan selection. However, these methods often suffer from long training times, limited adaptability across workloads, and high integration costs.

An alternative approach that preserves the existing optimizer architecture is Bao Marcus et al. (2021), which integrates a Tree Convolutional Neural Network Mou et al. (2016) with Thompson sampling to guide the selection of better hint sets. This hybrid approach improves plan generation without requiring a complete redesign of the optimizer, thereby balancing innovation with system compatibility.

Logical plan rewriting involves transforming a tree-structured representation of query operations, focusing on what to compute rather than how to compute it. This process is primarily driven by rule-based pattern matching, where heuristic rules—such as predicate push-down and operation merging—guide the transformation Levy et al. (1994); Pirahesh et al. (1992); Muralikrishna et al. (1992). Recent efforts aim to automate rule discovery. WeTune Wang et al. (2022) uses brute-force enumeration to identify and validate new rules, enhancing the internal optimizer's capabilities. QueryBooster Bai et al. (2023) introduces a connector for user-defined rules, enabling task-specific rewriting strategies. Traditional rule application orders are often fixed and suboptimal. LR Zhou et al. (2021) applies Monte Carlo Tree Search to explore effective rewriting sequences, while LLM-$R^2$ Li et al. (2024c) leverages large language models to recommend context-aware rewriting rules, improving adaptability and generalization.

End-to-end SQL rewriting aims to enhance transparency and usability by rewriting queries before they enter the DBMS pipeline. This approach enables holistic transformations and avoids the limitations of local rewriting efforts. Recent studies have explored the use of Large Language Models (LLMs) to facilitate this process. The DB-GPT framework Zhou et al. (2024) categorizes such approaches into three paradigms: in-context learning, LLM fine-tuning, and DB-specific pre-training. GenRewrite Liu & Mozafari (2024), a representative in-context learning method, designs prompts to

guide LLMs in SQL rewriting and stores generated natural language rules in the NLR2s repository for reuse. To mitigate LLM hallucinations, GenRewrite includes a validation and correction step to ensure the reliability of the rewritten queries.

Additionally, middleware-based rewriting has been explored to offload optimization tasks from the DBMS. This approach provides a flexible layer between the application and the database, enabling query transformation before execution Bai (2023). Similarly, query rewriting has been integrated into human-in-the-loop systems to support interactive data exploration, where users can iteratively refine queries based on intermediate results.

**Query Error Detection and Correction**    SQL query errors fall into two categories: syntax errors and semantic errors. Syntax errors occur when a query violates SQL's syntactical rules, preventing execution. Traditional debugging methods, as noted by Gathani et al. Gathani et al. (2020), lack automated correction and instead help users identify errors through techniques like visualizing intermediate results. Semantic errors arise when a query fails to return expected data, indicating a mismatch between the query's output and the user's intent. Verifying query equivalence is crucial in NL2SQL conversion Pourreza et al. (2024); Talaei et al. (2024); Gao et al. (2024c) and query rewriting models Dong et al. (2023); Liu & Mozafari (2024); Wang et al. (2022). Existing SQL equivalence provers use algebraic representations to verify query equivalence by solving mathematical problems Ding et al. (2023b); Zhou et al. (2019), offering high reliability but at high computational cost. For loosely bounded verification, heuristic rules and counterexample construction are employed Dong et al. (2023), while some studies leverage LLMs for reasoning and judgment Liu & Mozafari (2024).

---

**Algorithm 1:** Fragment Processing Strategy

---

**Input:** SQL query $Q$
**Output:** Analysis result set $S$
1  $S \leftarrow \emptyset$ ;                                                      // Initialize result set
2  $D \leftarrow \emptyset$
3  $Q_{main}, \{Q_{cte_j}\} \leftarrow \texttt{divide\_CTE}(Q)$
4  **if** $Q_{main} = \emptyset$ **then**
5  |    **return** $\emptyset$
6  **end**
7  $\{Q_{sub_i}\} \leftarrow \texttt{parse\_subqueries}(Q_{main})$
8  **foreach** $Q_{sub} \in \{Q_{sub_i}\}$ **do**
9  |    $S_{sub} \leftarrow \texttt{fragment\_processing}(Q_{sub})$
10 |    $S \leftarrow S \cup S_{sub}$
11 **end**
12 **foreach** $Q_{cte} \in \{Q_{cte_j}\}$ **do**
13 |    $S_{cte} \leftarrow \texttt{fragment\_processing}(Q_{cte})$
14 |    $S \leftarrow S \cup S_{cte}$
15 **end**
   // Main query analysis (details omitted)
16 **return** $S$

---

### A.4    DATASETS AND METRICS

**Datasets**.  To comprehensively evaluate the performance of SQLGOVERNOR, we select two representative benchmarks: BIRD-CRITIC Li et al. (2025) and BIRD Li et al. (2024b). Additionally, we have constructed a new dataset named Payment-SQL, which comprises analytical SQL queries derived from real industrial scenarios, specifically designed to evaluate performance in handling complex and diverse queries.

**BIRD-CRITIC** is an innovative SQL benchmark crafted to evaluate the critical capabilities of LLMs in diagnosing and resolving user issues within real-world database environments. The benchmark categorizes issues into four domains: *Query*, *Management*, *Personalization*, and *Efficiency*. These categories align with the core functionalities of SQLGOVERNOR. For our experiments, we utilize

a light version, `bird-critic-1.0-flash-exp`, which consists of 200 user issues on Post-greSQL.

**BIRD** serves as a challenging large-scale database text-to-SQL evaluation benchmark, designed to bridge the gap between academic research and practical applications. It encompasses 95 extensive databases and high-quality text-SQL pairs, with data storage reaching up to 33.4GB, spanning 37 professional fields. The validation set includes 1,534 test entries, offering a comprehensive evaluation of text-to-SQL translation capabilities. Notably, in utilizing this dataset, we employ SQLGO-VERNOR as a post-processing tool for NL2SQL models, aimed at further enhancing the quality of generated SQL queries.

**Payment-SQL dataset** originates from real-world industrial OLAP scenarios and is curated by human experts based on execution logs. It contains 50 SQL queries, each involving an average of 2 tables and 11 columns, drawn from a schema of 74 tables with thousands of fields. Designed specifically for evaluating SQL rewriting systems, Payment-SQL measures effectiveness through execution time comparisons before and after rewriting in the same environment—directly reflecting real-world performance gains. A key feature of Payment-SQL is its complexity: the average query length is 421 tokens , far exceeding that of BIRD's challenging category (107 tokens). According to Spider 2.0 Lei et al. (2024), where queries over 160 tokens are considered difficult, even the shortest query in Payment-SQL (173 tokens) qualifies as hard, with the longest reaching 1169 tokens. This makes Payment-SQL a rigorous and realistic benchmark for evaluating the robustness and scalability of SQL rewriting techniques in industrial applications. The dataset is available at `https://anonymous.4open.science/r/SQLGovernor-33DF`.

**Evaluation Metrics**. On the BIRD-CRITIC-FLASH dataset, we follow the official guidelines and use the success rate (SR) as the metric, as it effectively evaluates multiple aspects of performance due to the well-designed test cases. For the BIRD dataset, we employ both Execution Accuracy (EX) and Valid Efficiency Score (VES) metrics to comprehensively evaluate performance. In the case of the Payment-SQL dataset, rewriting effectiveness is assessed using Execution Time Saved (ETS) and Execution Time Optimization Gain (ETOG), calculated as follows:

$$\text{ETS} = \text{ET}_{\text{pre}} - \text{ET}_{\text{post}}, \ \text{ETOG} = \frac{\text{ETS}}{\text{ET}_{\text{pre}}} \times 100\%, \tag{4}$$

where $ET_{\text{pre}}$ represents the execution time before rewriting and $ET_{\text{post}}$ represents the execution time after rewriting. It is worth noting that when using ETS and ETOG to evaluate SQL rewriting tasks, we typically execute both the pre-optimized and post-optimized SQL queries in the same system while excluding interference factors such as execution caching to ensure the objectivity and reliability of the test results.

## A.5 MORE ABLATION STUDY

**Error Correction**    To evaluate the error correction capabilities of SQLGOVERNOR, we collect a set of syntactically and semantically incorrect SQL queries generated by two strong LLM-based NL2SQL systems—CodeS-7B and CodeS-15B—on the BIRD dataset. Queries that failed to execute due to syntax errors are fed into the Syntax Error Corrector, while those exhibiting semantic misalignment are routed to the Query Modifier for refinement.

Table 5 presents the results of the error correction capabilities in SQLGOVERNOR. The findings indicate that the module demonstrates strong error correction performance on the BIRD datasets, as evidenced by the predictive results from both baseline models. For the CodeS-7B model, we analyzed 691 erroneous cases, yielding an overall EX rate of 25.8%. Performance across difficulty levels shows EX rates of 26.4% for simple cases, 26.2% for moderate cases, and 22.0% for challenging cases. In contrast, the CodeS-15B model, evaluated on 667 erroneous cases, achieved an overall EX rate of 25.2%, with rates of 26.9% for simple cases, 23.1% for moderate cases, and 25.0% for challenging cases.

**Equivalence Verification**    We use the predictive results from the CodeS-7B model alongside golden SQL queries to establish positive and negative pairs. Correctly predicted SQL queries are classified as equivalent with the golden SQL (labeled as true), while incorrectly predicted queries are deemed nonequivalent (labeled as false). The results are presented in Table 6. We report two

Table 4: Evaluation on BIRD's dev set.

| Methods | Dev set | |
| --- | --- | --- |
| | EX(%) | VES(%) |
| **Prompt-based base models** | | |
| Codex Li et al. (2024b) | 34.35 | 43.41 |
| ChatGPT Li et al. (2024b) | 37.22 | 43.81 |
| GPT-4 Li et al. (2024b) | 46.35 | 49.77 |
| DIN-SQL + GPT-4 Pourreza & Rafiei (2024) | 50.72 | 58.79 |
| DAIL-SQL + GPT-4 Gao et al. (2024a) | 54.76 | 56.08 |
| **Fine-tuning-based base models** | | |
| T5-3B Li et al. (2024b) | 23.34 | 25.57 |
| CodeS-7B Li et al. (2024a) | 57.17 | 58.80 |
| CodeS-15B Li et al. (2024a) | 58.48 | 59.87 |
| XiYan-32B Gao et al. (2024b) | 67.01 | 67.79 |
| **With post-processing tools** | | |
| CodeS-7B + SQLFixAgent Cen et al. (2024) | 60.17 (↑3.00) | 63.15 (↑4.35) |
| CodeS-7B + SQLGOVERNOR | 64.02 (↑**6.85**) | 64.72 (↑5.92) |
| CodeS-15B + SQLGOVERNOR | 65.32 (↑6.84) | 67.87 (↑**8.00**) |
| XiYan-32B + SQLGOVERNOR | **68.97** (↑1.96) | **70.89** (↑3.10) |

Table 5: Error correction performance on the BIRD's dev set including syntactic and semantic levels.

| Error Data Statistics | Total | Simple | Mod. | Chall. |
| --- | --- | --- | --- | --- |
| **#CodeS-7B Error Case** | 691 | 333 | 267 | 91 |
| EX(%) | 25.8 | 26.4 | 26.2 | 22.0 |
| **#CodeS-15B Error Case** | 667 | 324 | 255 | 88 |
| EX(%) | 25.2 | 26.9 | 23.1 | 25.0 |

metrics-accuracy and F1 score-and present the results in Table 6, the overall accuracy for verification is 78.9% and F1 score is 79.3%, indicating effective performance of the Verifier. It is noteworthy that the scores for challenging queries are lower than those for simpler queries, which is expected given the increased complexity of the SQL statements.

Table 6: Equivalence verification performance on the predictive results of CodeS-7B.

| Data Category | Total | Simple | Mod. | Chall. |
| --- | --- | --- | --- | --- |
| **Verif. Accuracy(%)** | 78.9 | 81.2 | 76.9 | 71.0 |
| **Verif. F1(%)** | 79.3 | 84.3 | 69.9 | 57.1 |

### A.6 DETAILED EXPERIMENTAL ANALYSIS

**Capability in processing long and complex SQL** We analyze the capability of SQLGOVERNOR in handling long SQL queries from two common scenarios: error correction and rewriting. Figure 2a presents the detailed performance of SQLGOVERNOR when using CodeS-7B, CodeS-15B, and XiYan-32B as base models. The shaded bars illustrate the performance improvement achieved by SQLGOVERNOR over the base models. SQLGOVERNOR consistently outperforms the base models across all categories, with particularly notable gains in the Challenge SQL section of the BIRD dataset.

Furthermore, Table 2 illustrates the rewriting performance of SQLGOVERNOR on the industrial-level dataset Payment-SQL. Compared to general-purpose LLMs, SQLGOVERNOR exhibits a clear advantage in SQL rewriting tasks. Notably, the average token length of Payment-SQL reaches 421, far exceeding the complexity of SQL queries in the BIRD dataset. Additionally, all SQL queries in Payment-SQL meet the Hard SQL standard defined by Spider 2.0 Lei et al. (2024) (token length ¿

160). These results strongly demonstrate the superior rewriting capability of SQLGOVERNOR in handling long and complex SQL queries.

**Validation of productivity improvement**    To validate the effectiveness of SQLGOVERNOR in addressing productivity bottlenecks caused by fragmented SQL tool-chains, we conducted a controlled A/B testing with 60 practitioners from in-production data platform. Participants were stratified by SQL expertise (30 experts and 30 non-experts) and uniformly assigned to two groups-Group A: Utilizing the integrated SQLGOVERNOR framework; Group B: Operating equivalent discrete modules through manual orchestration. Each subject executed 50 standardized SQL governance tasks spanning evaluation, correction, rewriting, verification. We systematically measured-task completion time and tool-switching frequency. Results demonstrated statistically significant advantages for the integrated framework. Group A achieved 33% faster task completion, with non-experts exhibiting greater efficiency gains (41% improvement) compared to 25% for experts. This disparity correlates with Group B's tool-switching patterns, where practitioners incurred 18% temporal overhead reconstructing workflow contexts between discrete modules. The empirical evidence quantitatively confirms that SQLGOVERNOR's unified pipeline effectively mitigates fragmentation-induced productivity loss, particularly benefiting non-specialist users.

**"Evolving with every step"**    To validate the effectiveness of our expert-guided hybrid self-learning mechanism in continuously enhancing the performance of SQLGOVERNOR across various SQL-related tasks, we collect and retain results at different stages for an end-to-end task and an individual tasks. This approach allows us to assess how SQLGOVERNOR improves its capabilities through self-learning.

Specifically, for the end-to-end task, we use the predictive results of CodeS-15B on the BIRD's dev set. For SQL rewriting task, we choose the Payment-SQL dataset to examine the iterative gains of SQLGOVERNOR in long SQL rewriting scenarios. The experimental results shown in Figure 4 demonstrate that the hybrid self-learning approach not only enhances the performance of SQLGO-VERNOR but also provides a reliable foundation for its continuous rewriting in real-world industrial applications. Moreover, the effectiveness of this mechanism further validates the feasibility of transitioning from expert-centric knowledge base construction to an expert-guided hybrid self-learning framework, thereby providing methodological support for reducing the cost of complete reliance on experts for knowledge collection and maintenance.

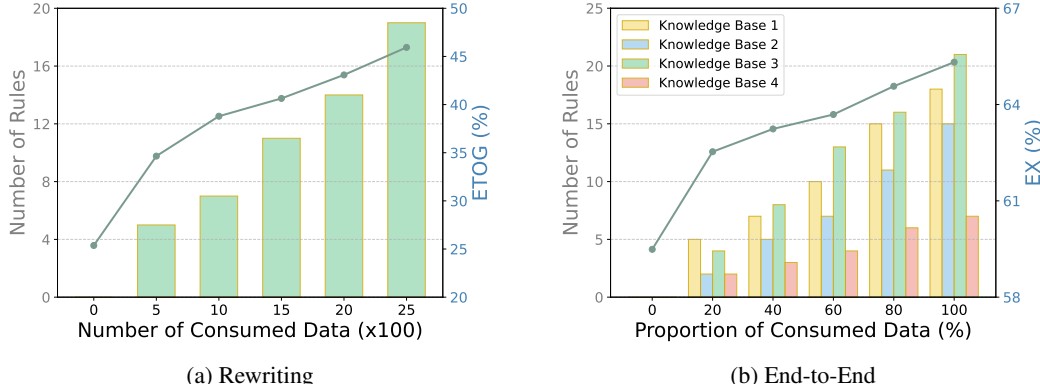

(a) Rewriting    (b) End-to-End

Figure 4: The performance metrics of SQLGOVERNOR across different stages of self-learning. The bar chart corresponds to the left y-axis, while the line chart corresponds to the right y-axis.

