# OpenReview forum: "SQLGovernor: An LLM-powered SQL Toolkit for Real World Application"
_ICLR.cc/2026/Conference — Submitted to ICLR 2026_

### Official Review · Reviewer_4rsk · 2025-10-27

**Soundness:** 3
**Presentation:** 2
**Contribution:** 2
**Rating:** 4
**Confidence:** 3

**Summary:**

The paper proposes SQLGovernor, an LLM-powered SQL toolkit designed for real-world OLAP applications. It unifies four functionalities—syntax correction, query rewriting, semantic modification, and equivalence verification—within a single framework supported by a knowledge management module. The system introduces a fragment-wise processing strategy for decomposing complex SQLs, and a hybrid self-learning mechanism that incrementally updates its rule base through DBMS log analysis and expert feedback. Experiments on BIRD, BIRD-CRITIC, and an internal Payment-SQL dataset show consistent improvements over baselines, with up to 10% gain in execution and verification metrics and notable runtime savings in production deployments.

**Strengths:**

- Provides a comprehensive, unified SQL toolkit that integrates multiple submodules and mitigates fragmentation in SQL processing pipelines.
- Demonstrates strong practical value with measurable performance gains and deployment evidence in real industrial environments.
- Uses realistic datasets (BIRD, BIRD-CRITIC, Payment-SQL) and well-structured evaluation protocols, illustrating both academic and applied significance.

**Weaknesses:**

- The paper’s learning component is limited—no parameter-level model training or end-to-end optimization is presented; improvements mainly stem from rule-based augmentation and prompt engineering.
- Computational overhead is nontrivial (up to 2× latency increase), which may hinder real-time or high-throughput applications.
- Generalization and theoretical insight are weak; the framework focuses on system integration rather than proposing new machine learning principles.
- Evaluation primarily covers SELECT-based OLAP queries; applicability to broader SQL tasks (e.g., transactional workloads, multi-schema joins) is unclear.
- Comparative scope is limited to open-source models (Qwen, CodeS, XiYan); no comparison against closed-source strong baselines (e.g., GPT-4, Claude).

This paper is technically solid and highly practical, but it falls short of ICLR’s focus on fundamental learning contributions. It would likely be more suitable for a **Demo or Industry Track at SIGMOD/VLDB/WWW**, unless the authors reframe the self-learning process as a genuine, learnable mechanism with quantitative evidence of adaptation.

**Questions:**

- How transferable are the learned rules across different DBMS engines or schema domains?
- Can the authors clarify whether the LLM ever adapts parameters through fine-tuning or reinforcement, or if all improvements are symbolic/knowledge-based?

---

### Official Review · Reviewer_2gqq · 2025-10-28

**Soundness:** 2
**Presentation:** 2
**Contribution:** 2
**Rating:** 2
**Confidence:** 4

**Summary:**

SQLGovernor is an LLM-powered SQL toolkit for OLAP workloads with four tools (syntax corrector, query rewriter, query modifier, equivalence verifier) backed by a knowledge management system. Key contributions: (1) fragment-wise processing that decomposes complex queries into subqueries for localized analysis, (2) hybrid self-learning that generates rules from execution logs with periodic expert validation, (3) unified framework integrating multiple SQL tasks. Evaluated on BIRD, BIRD-CRITIC, and an industrial dataset (Payment-SQL), showing 6-10% improvements over base models.

**Strengths:**

1. Real production deployment with industrial dataset: Payment-SQL (50 queries, avg 421 tokens, max 1169 tokens) represents actual OLAP complexity. System is deployed and improving over time

2. Practical hybrid approach to knowledge management: Combining LLM rule generation with periodic expert validation (every 2 weeks) is more scalable than pure manual curation. Automatic deduplication via embeddings + DBSCAN prevents rule bloat.

3. Strong empirical results on complex queries: Largest gains on "Challenging" category of BIRD (+8.28% for CodeS-15B). On Payment-SQL: 45.92% ETOG vs 31.25% for GenRewrite, 14.56% for Qwen2.5.

**Weaknesses:**

1. Lacks focused validation of individual contributions: Paper presents 4 tools + knowledge management + fragment processing + self-learning + intent classification but validates none in isolation. Which components actually contribute to the 6-10% gains? Without ablations, impossible to assess if any individual contribution is significant. Recommend narrowing scope to 2-3 core technical contributions with rigorous validation rather than presenting multiple components without understanding their individual impact.

2.  Missing baseline comparisons
 - No comparison with DBMS optimizers. PostgreSQL already does LEFT JOIN+NOT NULL→INNER JOIN  and IN(SELECT)→JOIN.
 - No comparison with simple GPT-4/Claude prompt + schema (the obvious baseline). The multi-stage pipeline might perform worse than one well-crafted prompt

3. Fragment processing: unvalidated core contribution
 - No ablation comparing WITH vs WITHOUT fragment processing on same queries
 - With modern long-context LLMs (Gemini 2.5 Flash 1M context), 421-token queries likely don't need decomposition
 - Fragment processing prevents cross-fragment optimizations (e.g., merging two subqueries scanning same table requires seeing both fragments together)

4. Query Modifier intent classification appears unnecessary
 - Classification accuracy: 78.9% with 21% error rate that propagates downstream
 - No ablation showing category-specific prompts outperform generic prompts
 - Modern LLMs (GPT-4, Claude) can infer modification type from request text - explicit pre-classification adds latency and error without proven benefit
 - Categories are limited (4 types, with "other" as catch-all)
 - Qwen3-32B direct classification: 84.3% accuracy vs embedding approach: 78.9% - why not just use LLM directly? Saving 0.18s on classification but losing 5.4% accuracy seems like bad trade-off that needs further justification

5. Correctness concerns with no mitigation strategy
 - Equivalence verifier: 78.9% accuracy means 21% of rewrites are incorrectly validated
 - No production safeguards discussed: dry-run testing, gradual rollout, result checksums, automatic rollback
 - Rules stored as vague natural language descriptions, not executable logic - inconsistent LLM interpretation
 - What happens when bad rewrites reach production?

6. Self-learning mechanism under-evaluated
 - Claims 25-35% cost reduction with no supporting evidence
 - Expert rejection rate not reported - if 80% of auto-generated rules are rejected, system just overwhelms experts
 - Rules are well-known patterns (LEFT JOIN→INNER, IN→JOIN, COUNT DISTINCT decomposition) documented since 1990s-2000s, not novel discoveries. Why not bootstrap from 30+ years of database research literature instead of rediscovering?

**Questions:**

Q1: Which components actually matter?
Please provide ablations isolating each major component's contribution: (a) fragment processing vs full-query, (b) intent classification vs direct LLM modification, (c) self-learning vs static expert rules, (d) knowledge retrieval vs no retrieval.

Q2: Why not use obvious baselines?
What's the performance of: (a) GPT-4 with simple one-shot prompt + schema vs your multi-stage pipeline, (b) well-tuned optimizer parameters?

Q3: Does fragment processing help modern LLMs?
With Gemini 2.0 Flash (1M context), GPT-5, or Claude 3.5, at what query length does fragment processing provide measurable benefit? Your queries average 421 tokens - well within single-pass capability.

Q4: How do you ensure correctness in production?
With 21% verifier error rate: (a) What safeguards prevent inefficient rewrites from reaching production? (b) Have you measured incidents of bad rewrites?

---

### Official Review · Reviewer_mZXb · 2025-11-02

**Soundness:** 2
**Presentation:** 3
**Contribution:** 2
**Rating:** 4
**Confidence:** 5

**Summary:**

This paper presents SQLGOVERNOR, an LLM-powered SQL toolkit designed for real-world analytical (OLAP) workloads. Unlike prior systems that focus narrowly on text-to-SQL or isolated SQL debugging tasks, SQLGOVERNOR proposes a unified multi-tool framework that includes syntax correction, query rewriting, semantic refinement, and consistency verification. The authors propose a fragment-wise processing strategy that recursively decomposes complex SQL into fragments (e.g., CTEs, subqueries) for localized optimization and correction, reducing the cognitive load on LLMs. The system also introduces an expert-guided hybrid self-learning mechanism, enabling rule extraction from DBMS logs and execution traces, validated and incorporated into a knowledge base over time. Experiments on BIRD, BIRD-CRITIC-Flash, and Payment-SQL workloads show performance improvements in accuracy and query efficiency.

**Strengths:**

S1. The problems addressed in the paper are well motivated in OLAP, especially the problems beyond NL2SQL.

S2. The fragments-based SQL decomposition with LLMs is reasonable and shows some promising results.

S3. The proposed system integrates post-processing pipeline for SQL beyond NL2SQL and self-improving rule generation grounded in DBMS feedback.

**Weaknesses:**

W1. The framework presents a broad toolkit, but the contributions are spread across many components. This makes the system feel somewhat fragmented, with each element receiving limited conceptual depth. A stronger anchoring around a single core technical innovation would significantly strengthen the work.

W2. The proposed solution does not provide any formal guarantee of semantic preservation for rewriting (LLM + heuristics only).

W3. The rule extraction stability and failure handling are not clearly explained. And knowledge base curation cost is not fully evaluated at scale. The authors should provide case studies of rule synthesis failures and mitigation.

W4. The performance overhead raises practicality concerns, particularly for real-time or latency-sensitive query systems. A more detailed latency breakdown per module, along with error analyses, would significantly strengthen the evaluation.

**Questions:**

Q1. How often does the rule-extraction process generate incorrect or redundant rules (false positives), and how costly is expert validation?

Q2. Can fragment processing harm optimization opportunities by breaking global context?

Q3. Does the toolkit support incremental feedback loops with execution feedback (runtime, cost estimates) beyond correctness logs?

Q4. When LLM rewrites harm performance and how does SQLGOVERNOR handle SQL dialects (e.g., Snowflake, Oracle analytic functions)?

Q5. What guardrails exist to prevent catastrophic LLM rewrites in production?

---

### Official Review · Reviewer_P6Ns · 2025-11-04

**Soundness:** 2
**Presentation:** 3
**Contribution:** 2
**Rating:** 4
**Confidence:** 4

**Summary:**

This paper proposes SQLGovernor, a unified LLM-powered toolkit for SQL governance (syntax correction, rewriting, modification, equivalence verification) with fragment-wise processing and a hybrid, expert-guided knowledge base. It shows consistent improvements on BIRD/BIRD-CRITIC and strong rewriting gains on an industrial Payment-SQL dataset.

**Strengths:**

S1. Practical unified toolkit targeting real OLAP issues; clear modularization (Corrector/Rewriter/Modifier/Verifier).
S2. Sensible design ideas: fragment-wise scoping, selective schema inclusion, knowledge retrieval.
S3. Empirical gains over strong baselines; Payment-SQL demonstrates large end-to-end efficiency benefits.

**Weaknesses:**

W1. Reproducibility and portability of industrial gains. Payment-SQL results are hard to reproduce (missing full env/specs, cold-cache controls, scripts); gains may depend on engine-specific hints/optimizers (e.g., MAPJOIN), with limited cross-DBMS evidence.

W2. Fragment processing underspecified; semantic safety not assured. Core partition/recomposition lacks rigorous handling of correlated subqueries, window frames, HAVING/ORDER/LIMIT, and CTE materialization; no systematic stress tests or guarantees that local fixes preserve global semantics.

W3. Equivalence verification too weak for safety-critical use. Limited to SELECT-based queries with ~78.9% accuracy; lacks hybrid fallback (SMT/constraint checks) on high-risk cases, error-mode analysis, and calibrated confidence/intervals.

W4. Insufficient attribution and ablations (incl. self-learning risks). No clean isolation of contributions (fragment vs whole-query; retrieval vs none; self-learning on/off); limited sensitivity/drift control for induced rules, weak reporting on expert veto/rollback; potential retrieval contamination not audited.

W5. Cost–quality and statistical rigor gaps. latency overhead without concrete mitigation (caching/pruning/fast paths) or Pareto analysis; baseline fairness/configs under-specified; improvements lack error bars, significance tests, or variability across seeds.

**Questions:**

1. Can you release full Payment-SQL replication assets (DDL, data mock or generators, execution scripts, engine configs) and show cross-DBMS results (e.g., Postgres/StarRocks/SparkSQL) to establish portability?
2. How do you guarantee semantic preservation with fragments in the presence of correlated subqueries, window frames, and HAVING/ORDER/LIMIT interactions? Any formal invariants or comprehensive stress suite?
3. Can you add a hybrid verifier: selective SMT/constraint checks on uncertain or high-impact rewrites, with calibrated thresholds to contain risk?
4. Please provide an ablation that toggles: {fragment on/off} × {knowledge retrieval on/off} × {self-learning on/off}, with statistical significance and error bars.
5. What mechanisms prevent knowledge drift/bloat (misinduced rules)? Report expert veto rates, rollback policies, and rule lifecycle metrics.
6. Can you present a cost–quality Pareto curve (latency vs EX/VES/SR) and concrete latency reductions via caching/pruning/fast-paths?

---

### Meta-Review · Area_Chair_gXRj · 2026-01-03

**Summary:**

Reviewers criticized the framework's broad but shallow contributions without isolated ablations for components like fragment processing, self-learning, and knowledge retrieval; missing comparisons to strong baselines such as DBMS optimizers, GPT-4, or Claude; underspecified semantic preservation in fragment handling, weak equivalence verification with high error rates, reproducibility gaps in industrial evaluations, and significant latency overheads without mitigations, informing a suggested rejection due to limited novelty, rigor, and alignment with ICLR's focus on fundamental ML advancements.

**Reviewer Concerns:**

No rebuttal is present in the document, so none of the concerns—such as ablations, baselines, semantic safety, verification accuracy, reproducibility, and overhead—have been addressed; all remain outstanding.

**Reviewer Scores:**

Without a rebuttal or additional discussion, reviewers would likely retain their original scores, with P6Ns, mZXb, and 4rsk at 4 (marginally below), 2gqq at 2 (reject).

---

### Decision · Program_Chairs · 2026-01-26

Reject